# Development of an Optimized MALDI-TOF-MS Method for High-Throughput Identification of High-Molecular-Weight Glutenin Subunits in Wheat

**DOI:** 10.3390/molecules25184347

**Published:** 2020-09-22

**Authors:** You-Ran Jang, Kyoungwon Cho, Se Won Kim, Susan B. Altenbach, Sun-Hyung Lim, Jae-Ryeong Sim, Jong-Yeol Lee

**Affiliations:** 1National Institute of Agricultural Science, RDA, Jeonju 54874, Korea; jang6122@korea.kr (Y.-R.J.); sewonk@korea.kr (S.W.K.); lego0815@korea.kr (J.-R.S.); 2Department of Biotechnology, College of Agriculture and Life Sciences, Chonnam National University, Gwangju 500-757, Korea; kw.cho253@gmail.com; 3USDA-ARS, Western Regional Research Center, 800 Buchanan Street, Albany, CA 94710, USA; Susan.Altenbach@ARS.USDA.GOV; 4Division of Horticultural Biotechnology, Hankyong National University, Anseong 17579, Korea; limsh2@hknu.ac.kr

**Keywords:** high-molecular-weight glutenin subunits, MALDI-TOF-MS, wheat flour quality

## Abstract

Because high-molecular-weight glutenin subunits (HMW-GS) are important contributors to wheat end-use quality, there is a need for high-throughput identification of HMW-GS in wheat genetic resources and breeding lines. We developed an optimized method using matrix-assisted laser desorption/ionization time-of-flight mass spectrometry (MALDI-TOF-MS) to distinguish individual HMW-GS by considering the effects of the alkylating reagent in protein extraction, solvent components, dissolving volume, and matrix II components. Using the optimized method, 18 of 22 HMW-GS were successfully identified in standard wheat cultivars by differences in molecular weights or by their associations with other tightly linked subunits. Interestingly, 1Bx7 subunits were divided into 1Bx7 group 1 and 1Bx7 group 2 proteins with molecular weights of about 82,400 and 83,000 Da, respectively. Cultivars containing the 1Bx7 group 2 proteins were distinguished from those containing 1Bx7^OE^ using well-known DNA markers. HMW-GS 1Ax2* and 1Bx6 and 1By8 and 1By8*, which are difficult to distinguish due to very similar molecular weights, were easily identified using RP-HPLC. To validate the method, HMW-GS from 38 Korean wheat varieties previously evaluated by SDS-PAGE combined with RP-HPLC were analyzed by MALDI-TOF-MS. The optimized MALDI-TOF-MS method will be a rapid, high-throughput tool for selecting lines containing desirable HMW-GS for breeding efforts.

## 1. Introduction

The glutenins are abundant storage proteins accumulated in developing wheat endosperm that consist of high-molecular-glutenin subunits (HMW-GS) and low-molecular-weight subunits (LMW-GS) linked by disulfide bonds into large protein polymers. These polymers contribute to the viscoelastic properties of wheat flour dough in bread-making.

The HMW-GS are encoded at the *Glu-A1*, *Glu-B1*, and *Glu-D1* loci on the long arms of chromosomes 1A, 1B, and 1D, respectively [1]. Each locus contains tightly-linked genes encoding two different types of HMW-GS, referred to as x- and y-type subunits [2,3]. The x-type subunits generally have higher molecular weights (MWs) than the y-type subunits. Most common wheat varieties express only three to five subunits due to the silencing of some genes, particularly those encoding the y subunits at the *Glu-A1* locus [4]. Each subunit has been assigned a unique number and been given different quality scores related to bread-making properties [1,5,6,7]. For instance, the allelic pair 1Dx5 + 1Dy10 is associated with superior bread-making quality, especially dough strength. 1Ax2* and 1Dx17 + 1Dy18 also have positive effects on bread-making quality [8,9], while the allelic pair 1Dx2 + 1Dy12 is associated with poor bread-making quality [1,4,6]. In addition, overexpression of 1Bx7 subunit (1Bx7^OE^) in wheat with two functional copies of the 1Bx7 gene correlates with good dough strength [7,10,11,12,13]. Wheat varieties with different HMW-GS compositions are critical for providing the strength and extensibility of flour dough for different end uses [14].

In the past decades, a variety of techniques have been developed to characterize the allelic variation of HMW-GS. These include sodium dodecyl sulfate-polyacrylamide gel electrophoresis (SDS-PAGE), two-dimensional electrophoresis (2-DE), polymerase chain reaction (PCR), high-performance capillary electrophoresis (HPCE), reversed-phase high-performance liquid chromatography (RP-HPLC), and matrix-assisted laser desorption/ionization time-of-flight mass spectrometry (MALDI-TOF-MS) [15,16,17,18,19,20,21,22,23]. SDS-PAGE is the method that is most frequently used in breeding programs because it is simple and easy to perform [20,24]. However, SDS-PAGE has poor reproducibility compared to other methods, and HMW-GS with similar electrophoretic mobilities are difficult to identify. 2-DE and tandem mass spectrometry techniques were also developed but their applications are limited because of the technical expertise required and high cost. PCR is a simpler technique and currently used in many breeding programs [25,26,27,28,29]. However, primers are not available for all HMW-GS alleles, particularly those identified recently. HPCE satisfied the requirements of efficient, fast, and automated separation [30,31,32]. However, it is difficult to achieve both high resolution and reproducibility and some HMW-GS yield multiple peaks [20,22]. RP-HPLC has been used generally to select specific subunits associated with superior quality [16,17,19,20,21,33]. In RP-HPLC, subunits are separated according to their surface hydrophobicities [34] and subunits with higher hydrophobicities elute faster than subunits with lower hydrophobicities. Compared with other methods, RP-HPLC is convenient, extremely precise, fully automated, and easy to use for identifying many HMW-GS. However, HMW-GS having similar hydrophobicities cannot be differentiated by this method. RP-HPLC also requires long separation times and consumes an abundance of chemicals. As a result, RP-HPLC is not available to many breeding programs. MALDI-TOF-MS overcomes many of the limitations of other methods and has become a powerful tool for characterizing wheat storage proteins because of its accuracy and sensitivity [35,36,37,38,39,40]. It also requires relatively small samples of protein and only a few minutes per sample for analysis [37].

In this study, we established optimal conditions for the separation of HMW-GS by MALDI-TOF-MS. Twenty-two HMW-GS of bread wheat were distinguished successfully in standard wheat cultivars. The method was then applied to 38 varieties of Korean wheat to determine HMW-GS compositions.

## 2. Results and Discussion

### 2.1. Sample Optimization for MALDI-TOF-MS Analysis

To develop a reliable method for the resolution of HMW-GS by MALDI-TOF-MS, four different experimental parameters were optimized using flour from the wheat cultivar “Chinese Spring”. These included the use of an alkylating reagent during protein extraction, four different solvent components, three different dissolving volumes, and four different matrix II components (Figure 1, Table 1). In total, 96 experiments with different combinations were performed. Conditions that yielded the best spectra are shown in Figure 1.

#### 2.1.1. Treatment of 4-Vinylpyridine as Alkylating Reagent in Protein Extraction

In the glutenin extraction step, 4-vinylpyridine (4-vp) was added to alkylate the cysteine residues and stop the formation of disulfide bonds [41]. 4-vp-treated samples showed increased resolution of mass spectra when compared to non-treated samples (Figure 1A). The molecular weights determined by MALDI-TOF-MS for 4-vp-treated or non-treated HMW-GS are shown in Appendix A. The increase in molecular mass of individual HMW-GS 1Dy12, 1By8, 1Bx7, and 1Dx2 in 4-vp treated samples was 737.2, 714.9, 401.6, and 425.7 Da, respectively, which is in good agreement with previous knowledge that y-type HMW-GS have 7 cysteine residues, and most x-type HMW-GS have 4 cysteine residues (with the exception of 1Dx5 and 1Bx20 that have 5 and 2 cysteines, respectively). Although the measured increase in molecular mass differs somewhat from the 4-vp molecular mass of 105.14 Da, it is within the instrument’s common error range of a 90–115 Da increase for one cysteine [42]. Taking together the results of Wang et al. [42] and our results in this study, treatment with 4-vp showed enhanced resolution of mass spectra and could give useful information about the numbers of cysteine residues in the HMW-GS.

#### 2.1.2. Solvent Components

HMW-GS are difficult to ionize because of their large molecular masses. Four different solvents for HMW-GS were evaluated, including TA30 (30% acetonitrile (ACN), 0.1% trifluoroacetic acid (TFA)), TA50 (50% ACN, 0.1% TFA), SA in TA30 (10 μg/μL sinapinic acid (SA) in TA30), and SA in TA50 (10 μg/μL SA in TA50) (Figure 1B). SA was added to the solvent to increase the ionization and is well known to be useful for the analysis of higher mass proteins [43]. The best resolution of the alkylated protein sample was obtained with SA in TA30 (Figure 1B).

#### 2.1.3. Dissolving Volume

To test dilution volumes, 1 μL sample was diluted to 10 μL, 50 μL, and 100 μL of the resolving solvents. Figure 1C shows the results when the sample was diluted with SA in TA30. The 10 μL sample did not perform properly because the protein concentration was too high. The best results were obtained with a dilution volume of 50 μL.

#### 2.1.4. Matrix II Components

The matrix transfers the energy required for ionization from the laser to the sample molecules. Two matrixes were prepared according to the double layer method using SA recommended by manufacturer. SA is considered best matrix to analyze high molecular mass proteins over 5 kDa. Matrix I was SA in EtOH at a concentration of 10 mg/500 μL as recommended by the manufacturer’s manual. Four different matrix II compositions—SA in 30% ACN, SA in 50% ACN, SA in TA30, and SA in TA50—were tested and compared in Figure 1D. The addition of TFA to the matrix solution provides high intense peaks [43]. SA in TA50 was the most suitable as matrix II (Figure 1D).

As a result, in the optimized method alkylated proteins were dissolved in 50 μL SA in TA30. SA in TA50 was used for Matrix II.

### 2.2. Identification of HMW-GS Compositions in Standard Wheat Varieties

The HMW-GS compositions of 24 standard wheat varieties that have been well studied in various references were characterized by MALDI-TOF-MS using the optimized method [20,33,44,45,46,47,48,49]. Allelic variation of standard wheat varieties at *Glu-A1*, *Glu-B1*, and *Glu-D1* includes 22 subunits: 1Ax1 and 1Ax2* at *Glu-A1*, 1Bx6, 1Bx7, 1Bx7^OE^, 1Bx13, 1Bx14, 1Bx17, 1Bx20, 1By8, 1By8*, 1By9, 1By15, 1By16, 1By18, 1By20 at *Glu-B1* and 1Dx2, 1Dx2.2, 1Dx4, 1Dx5, 1Dy10, and 1Dy12 at *Glu-D1* (Table 2). The MALDI-TOF-MS analysis of each variety was performed five times. The mass spectra of the HMW-GS for standard wheat cultivars showed 3–5 distinct peaks with good resolution which allowed accurate molecular masses to be obtained (Figure 2). Average molecular weights (MW) and relative standard deviations (RSD) are shown in Table 2. RSDs for all subunits were less than 0.092, indicating that the molecular masses for all subunits were highly reproducible within the experiment.

Molecular weights of HMW-GS determined by MALDI-TOF-MS were compared with those determined from mature protein sequences deduced from HMW-GS genes with the exception of 1By8* and 1Dx4 for which gene sequences were not available in public databases (Table 3). N-termini of proteins were as determined by Shewry et al. [52] and the molecular weight of 4-vp (105.14 Da for each cysteine residue) was deducted from molecular weights determined by MALDI-TOF-MS. Predicted and measured molecular weights differed by less than 230 Da with errors less than −0.30% for all HMW-GS except 1Bx7, 1Bx17, and 1Dx2.2. Considering the error rate of our mass spectrometer, this indicates that our optimized method for sample extraction and spectral analysis is reliable. In comparison, 32 of 94 HMW-GS measured by Gao et al. [20] differed by more than 230 Da from their predicted molecular weights. It is more difficult to compare the accuracy of measurements in the Liu et al. [50] study since the measured values were presented as an average for the protein type and rounded to the nearest hundred. Nonetheless, values obtained for Bx14, By15, and Dy12 differed by more than 352 from molecular weights predicted from gene sequences. As more HMW-GS gene sequences have become available, small differences in the sequences of genes from different cultivars have become apparent, resulting in minor changes in the molecular weights of the encoded proteins. Nonetheless, in the current study the same subunit generally displayed similar molecular masses in different genotypes. However, two distinct molecular weights with a difference of ~600 Da were detected in genotypes containing the 1Bx7 HMW-GS, 82,400 Da (group 1) and 83,000 Da (group 2) (Table 3). Interestingly, the molecular masses detected in genotypes containing 1Bx7^OE^ were also 83,000 Da. In previous studies, it was not possible to discriminate 1Bx7 and 1Bx7^OE^ by either mobility in SDS-PAGE or retention time in RP-HPLC [33]. A molecular weight difference of about 600 Da suggests that the group 2 proteins may contain an additional six amino acid residues. Butow et al. [10] revealed that compared to 1Bx7, 1Bx7^OE^ has an 18 bp insertion in the repetitive domain of its coding sequence, resulting in an additional six amino acids near the C-terminal end. To further investigate the observed differences in the 1Bx7 subunits among cultivars, we performed PCR using DNA primers from Ma et al. [53] that span the region of the 18 bp insertion in 1Bx7^OE^ (Figure 3). Apexal, Cheyenne, Neepawa, Norin 61, Orca and Soissons each yielded two fragments of 650 bp and 750 bp, while the amplification products in Cappelle-Despre, Chinese Spring, Nanbu-Komugi, Orepi, and Petrel were slightly larger: ~670 bp and 770 bp (Figure 3A). In addition, we used primers developed by Ragupathy et al. [54] that detect the gene duplication found in cultivars containing 1Bx7^OE^ (Figure 3B,C). These primers amplified only Glenlea and IT166460 that are known to contain the duplicated gene. Our results suggest that there are two groups of similar 1Bx7 proteins that differ in molecular weight by about 600 Da. The group with the higher molecular weight is divided further into 1Bx7 group 2 and 1Bx7^OE^ that differ by the presence of a gene duplication.

In this study, HMW-GS 1Dx 2.2 with a predicted molecular weight over 100 kD was identified in the Japanese wheat Norin 61 and the Korean wheat Baekjoong. Although the mass range of the instrument used in this study was set to 60,000–110,000 Da, the difference between the measured molecular weight (100,390 Da for Norin 61 and 100,388 Da for Baekjoong) and the calculated molecular weight (100,866 Da) was −0.49%. It is estimated that the measurement error was due to the high molecular mass of the proteins. 1Dx2.2 was either not identified in other MALDI-TOF-MS studies [23,50] or the measured molecular weights had much higher errors [20]. For example, in the study of Gao et al. [20], the measured molecular weight of 1Dx2.2 was only 86,340 Da.

The precision of MALDI-TOF-MS measurements in the Gao et al. [20] study and the current study can be compared in cases where HMW-GS gene sequences are available from the same cultivar that was analyzed in both studies. The molecular weights of the mature Bx7, By8, Dx2, and Dy12 proteins deduced from gene sequences from Chinese Spring are 83,123, 75,131, 87,105, and 68,528, respectively (Huo et al., in preparation). The measured molecular weights of these proteins in the Gao et al. [20] study were 82,749, 75,488, 87,120, and 68,935, respectively, differing by more than 357 Da for three of the four subunits. The molecular weights measured for these HMW-GS in the current study were 83,067, 75,150, 87,072, and 68,674, differing by only 19 to 146 Da. Gene sequences for Bx14 (KF733216) and By15 (KF733215) from the cultivar Hanno have also been reported. The predicted molecular weights determined from the gene sequences were 82,343 for Bx14 and 74,739 for By15. In the Gao et al. [20] study, the measured molecular weights of Bx14 and By15 from Hanno were 82,505 and 75,282, differing by 162 and 543 Da, respectively, while those obtained in the current study were 82,365 and 74,790, differing by less than 51 Da.

Figure 4 summarizes the average MWs and the margins of error for each subunit. 1Ax1, 1Bx14, 1Bx17, 1Bx20, 1By9, 1By16, 1Dx2, 1Dx2.2, 1Dx4, 1Dx5, 1Dy10, and 1Dy12 differ by more than 500 Da and can be distinguished easily. Subunits that differ in MWs by less than 500 Da—1Ax2*, 1Bx6 and 1Bx7, 1Bx7 ^OE^, 1Bx13 and 1By8, 1By8*, 1By15, 1By18, and 1By20—cannot not be differentiated. However, 1Bx13, 1By15, 1By18, and 1By20 could be identified by considering the sizes of their tightly linked pairs, 1By16, 1Bx14, 1Bx17, and 1Bx20, respectively. Identification of other subunits is difficult because they have very similar molecular weights. 1Ax2 * and 1Bx6 have average molecular weights of 86,594 Da and 86,843 Da, respectively, which differ by only 249 Da (Table 2; Figure 4). 1By8 and 1By8* with average MWs of 75,874 Da and 75,966 Da, respectively, also are difficult to distinguish because they differ by only 92 Da (Table 2; Figure 4). However, both 1Ax2* and 1Bx6 and 1By8 and 1By8* can be distinguished easily by RP-HPLC as shown in Figure 5 and our previous study [33].

### 2.3. Identification of HMW-GS in 38 Korean Wheat Cultivars by MALDI-TOF-MS

To determine the feasibility of the methods developed in this study for use in breeding programs, the optimized MALDI-TOF-MS method was used to assess the HMW-GS compositions of 38 Korean wheat lines (Table 4, Appendix A). MALDI-TOF analyses were performed five times for each variety and the average MWs and RSDs of HMW-GS from these analyses are shown in Table 4. The RSDs ranged from 0.011 to 0.096, indicating that the identification of HMW-GS by our MALDI-TOF-MS method is very reliable. In order to identify HMW-GS of each variety, first of all, the molecular weight measured by MALDI-TOF-MS was compared with those of the box and whisker plots of standard wheat varieties in Figure 4. Subunits not distinguished by apparent molecular weight differences were judged by tightly linked subunit pairs or by RP-HPLC as described above for 1Ax2* and 1Bx6 and 1By8 and 1By8*. The identified HMW-GS of each variety in this study was exactly in agreement with the results obtained using RP-HPLC combined with SDS-PAGE, a far more time-consuming method (Table 4; Figure 6) [33].

Interestingly, 29 out of 38 Korean wheat varieties (76.3%) contained 1Bx7. As shown in Appendix A, after subtraction of the molecular weight of 4-vp, molecular weights of proteins determined by MALDI-TOF were approximately 82,400 in 16 varieties (1Bx7 group 1) and ~83,000 Da in 13 varieties (1Bx7 group 2). PCR analysis using primers of [53] revealed larger amplification products for the group 2 varieties, suggesting that these 1Bx7 group 2 genes contain the 18 bp insertion (Appendix A). Indeed, recent DNA sequence analysis confirmed that the 1Bx7 genes in the group 2 cultivars Keumkang and Olgeuru were identical to 1Bx7^OE^ gene, EU157184 (unpublished observations). However, PCR analysis with the primers of [54] indicated that none of the Korean lines contain the gene duplication (Appendix A). Additionally, it is notable that a number of other HMW-GS were not found among the 38 Korean wheat varieties. These include 1Bx6, 1Bx14, 1Bx20, 1By15, 1By20, and 1Dx4.

## 3. Materials and Methods

### 3.1. Plant Materials

The standard wheat (*Triticum aestivum* L.) cultivars for HMW-GS analysis were kindly provided from the National Bioresource Project-Wheat (NBRP-Wheat, https://shigen.nig.ac.jp/wheat/komugi/) in Japan and The U.S. National Plant Germplasm System (NPGS, https://www.ars-grin.gov/npgs/) in USA as listed in Table 2. Grain of 38 Korean hexaploid wheat cultivars was harvested in 2016 by RDA National Institute of Crop Sciences, Jeonju, Korea.

### 3.2. Glutenin Extraction

Glutenin was prepared by the modified method of Singh et al. [41]. Flour (30 mg) was extracted with 1 mL of 50% (*v*/*v*) propanol at 65 °C for 30 min to remove gliadins. After centrifugation at 10,000 *g* for 10 min, the supernatant was removed. These steps were repeated two times with the resulting pellet to remove gliadin completely. The resulting pellet was then extracted with 150 μL of extraction buffer (50% (*v*/*v*) propanol, 0.08 M Tris-HCl (pH 8.0) containing 1% (*w*/*v*) dithiothreitol (DTT)) at 65 °C for 30 min followed by centrifugation at 10,000 *g* for 5 min. Glutenins in the supernatant fraction were precipitated with 100 μL of cold acetone. Alternately, cysteine residues in the glutenins were alkylated by the addition of 150 μL of extraction buffer containing 1.4% 4-vinylpyridine (*v*/*v*) to the supernatant fraction and incubation at 65 °C for 15 min. After centrifuging at 10,000 *g* for 2 min, 200 μL of supernatant was precipitated with 135 μL of cold acetone and stored at −20 °C. Before MALDI-TOF-MS analysis, pellets were washed with cold acetone and dried at RT.

### 3.3. Sample Optimization

To optimize MALDI-TOF-MS resolution, solvent components, dissolving volume, and matrix II components were considered in addition to the effects of 4-vp on sample preparation (Table 1). Solvent components evaluated were (1) 30% acetonitrile (ACN, *v*/*v*) containing 0.1% trifluoroacetic acid (TFA) referred to as TA30, (2) 50% ACN (*v*/*v*) containing 0.1% TFA referred to as TA50, (3) sinapinic acid (SA) dissolved in 30% ACN (*v*/*v*) containing 0.1% TFA at a concentration of 10 μg/μL referred to as SA in TA30, and (4) SA dissolved in 0.1% TFA in 50% ACN (*v*/*v*) at a concentration of 10 μg/μL referred to as SA in TA50. After choosing the optimized composition of solvent components, the three dissolving volumes of 10 μL, 50 μL, and 100 μL were compared. Finally, four Matrix II solutions were investigated in which SA was dissolved at a concentration of 20 μg/μL in (1) 30% ACN (*v*/*v*); (2) 50% ACN (*v*/*v*); (3) 30% ACN (*v*/*v*) containing in 0.1% TFA; and (4) 30% ACN (*v*/*v*) containing in 0.1% TFA.

### 3.4. MALDI-TOF-MS

HMW-GS were analyzed by MALDI Microflex LT (Bruker Daltonics, Bremen, Germany) equipped with a 60 Hz nitrogen laser. The parameters of the instrument were used with the following settings; mass range: 60,000 to 110,000 Da, sample rate: 1.00 GS/s, laser shots: 100, laser power: 85%, laser frequency: 80, and detector gain: 13.3X. Spectra were obtained in positive ion mode. Bovine serum albumin (66,463 Da) was used as external standard for mass assignment.

### 3.5. RP-HPLC

The analysis of HMW-GS by RP-HPLC was described by Jang et al. [33]. RP-HPLC of HMW-GS was performed on a Waters Alliance e2695 equipped with Agilent ZORBAX 300SB-C_18_ column (5 μm, 4.6 × 250 mm i.d., Agilent Technologies, Santa Clara, CA, USA). The solvents (A) water and (B) ACN, both containing 0.1% (*v*/*v*) TFA were used as the mobile phase. Before use, solvents A and B were degassed for 30 min. Glutenin pellets were washed with 0.07% DTT in acetone two times and then dried. Dried glutenin pellets were dissolved completely in 500 μL of 0.1% TFA in 20% ACN and filtered using a 0.45 μm, Whatman PVDF syringe filter. Ten μL of each sample was injected and eluted at 0.8 mL/min using a linear gradient of 23–44% of solvent B for 70 min. The RP-HPLC analysis of HMW-GS was carried out at 60 °C column temperature and monitored at a wavelength of 206 nm.

### 3.6. Wheat DNA Extraction and PCR Analysis

Genomic DNA was extracted from 50 mg of wheat flour for the standard cultivars using a DNeasy^®^ Plant Mini Kit (Qiagen, Hilden, Germany) following the manufacturer’s instructions. Genomic DNA was quantified using a NanoDrop spectrophotometer (Thermo Scientific, Waltham, MS, USA) and diluted to 50 ng/μL. PCR was performed in a reaction volume of 25 μL using 200 ng of genomic DNA, 1.25 U of GoTaq DNA polymerase (Promega, Madison, WI, USA), 1x Green GoTaq reaction buffer (containing 1.5 mM MgCl_2_), 200 μM of dNTP mix (Bioneer, Daejeon, South Korea) and 10 pmole of forward and reverse primers. Bx7 coding region primers were forward 5′-CGCAACAGCCAGGACAATT-3′, and reverse 5′-AGAGTTCTATCACTGCCTGGT-3′ [53]. Left junction primers were: forward 5′-ACGTGTCCAAGCTTTGGTTC-3′ and reverse 5′-GATTGGTGGGTGGATACAGG-3′; and right junction primers were forward 5′-CCACTTCCAAGGTGGGACTA-3′ and reverse 5′-TGCCAACACAAAAGAAGCTG-3′ [54]. Amplification conditions for the PCR reaction were an initial cycle at 95 °C for 5 min, followed by 34 cycles of 94 °C for 30 s, 57 °C for 30 s, and 72 °C for 2 min, followed by a final extension at 72 °C for 5.25 min. PCR products were resolved on 1.5–2.0% agarose gels in 0.5x TBE buffer, stained with ethidium bromide and visualized under UV.

## 4. Conclusions

MALDI-TOF-MS is an accurate and high-throughput method for determining the HMW-GS composition of wheat varieties. Analyses can be completed in about one minute per sample and require only minimal amounts of protein and solvents. Eighteen HMW-GS in 24 standard wheat varieties grown in countries throughout the world were distinguished using the method optimized in this study. Additionally, varieties previously found to have the 1Bx7 subunit were shown to contain one of two different subunits that differed by 600 Da in molecular weight. The optimized method was further used to identify 15 different HMW-GS in 38 Korean wheat cultivars and will be valuable for the future analysis of large numbers of wheat breeding lines.

## Figures and Tables

**Figure 1 molecules-25-04347-f001:**
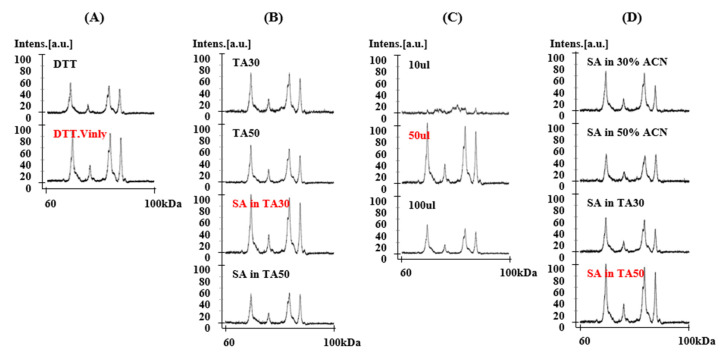
Effects of four major factors on matrix-assisted laser desorption/ionization time-of-flight mass spectrometry (MALDI-TOF-MS) resolution of Chinese Spring using MALDI-TOF-MS. (**A**) treatment with alkylation reagent in glutenin extraction, (**B**) solvent components, (**C**) dilution volume, and (**D**) matrix II components. Parameters shown in red were found to be optimal and were incorporated into the new method.

**Figure 2 molecules-25-04347-f002:**
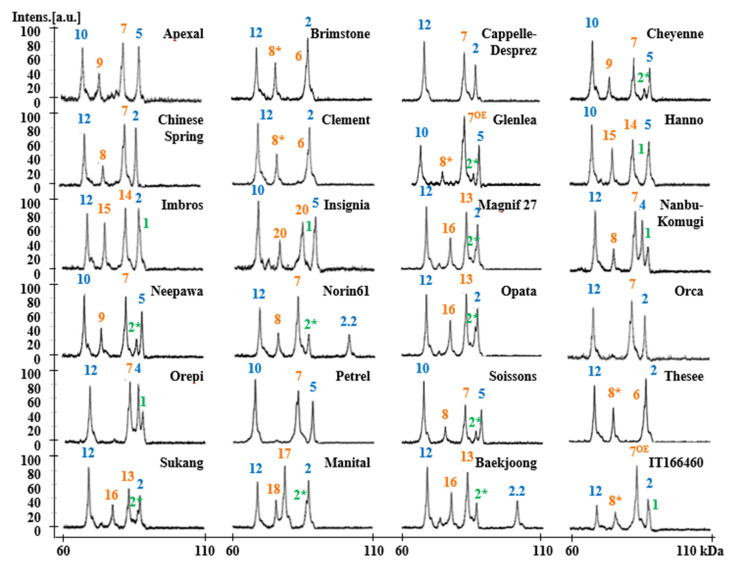
HMW-GS profiles of standard wheat cultivars determined by MALDI-TOF-MS. Subunits corresponding to chromosomes 1A, 1B, and 1D are displayed in green, orange, and blue, respectively.

**Figure 3 molecules-25-04347-f003:**
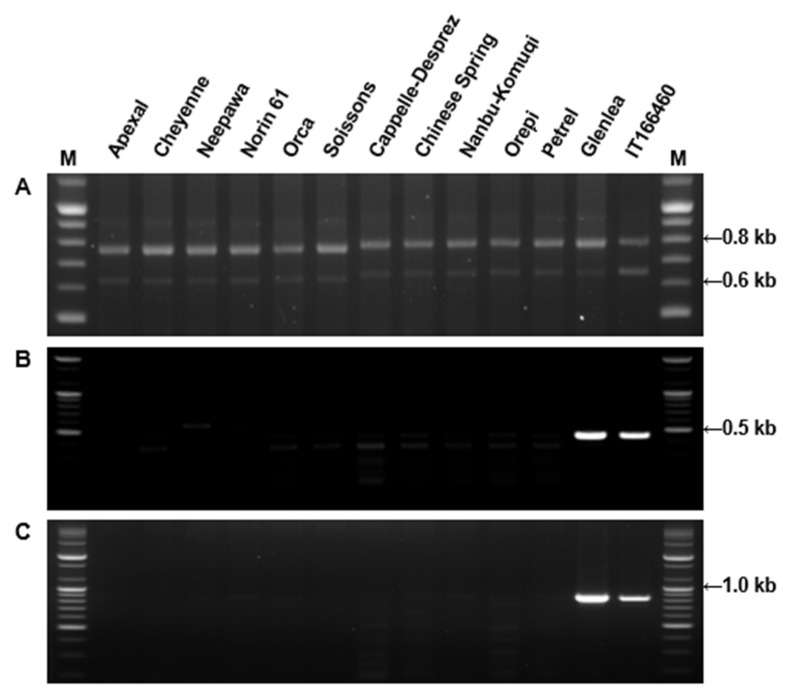
Amplification of DNA from standard cultivars containing 1Bx7 or 1Bx7OE genes. (**A**) PCR primers span the region of the 18 bp insertion in 1Bx7OE; (**B**) PCR primers from the left junction of the retroelement and duplicated region of Bx7OE; (**C**) PCR primers from right junction of the retroelement and duplicated region of Bx7OE. Glenlea and IT166460 were used as positive controls for Bx7OE. The 100 bp Plus DNA Ladder is shown in lanes (**M**).

**Figure 4 molecules-25-04347-f004:**
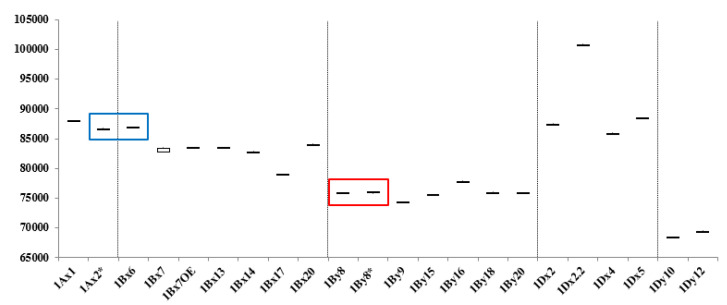
Box and whisker plot of molecular weights of individual subunits in standard wheat cultivars. Red and blue boxes indicate subunits differing by less than 300 Da that were not distinguished by MALDI-TOF-MS, but could be differentiated by RP-HPLC.

**Figure 5 molecules-25-04347-f005:**
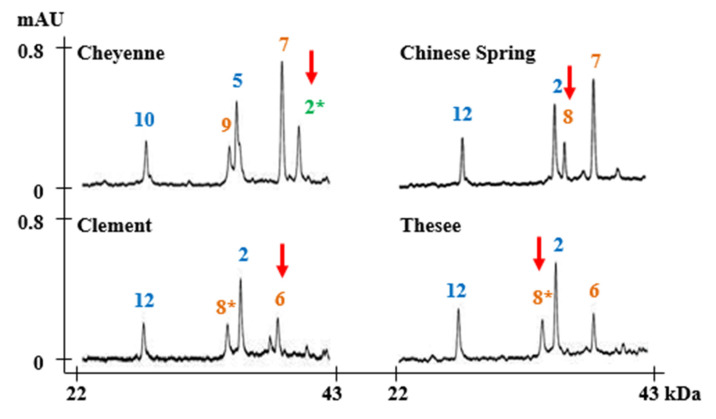
RP-HPLC analysis of subunits that could not be distinguished by MALDI-TOF-MS. Subunits corresponding to chromosomes 1A, 1B, and 1D are displayed in green, orange, and blue, respectively. Unidentified subunits 1Ax2*, 1Bx6 and 1Bx8, and 1Bx8* are indicated with red arrows.

**Figure 6 molecules-25-04347-f006:**
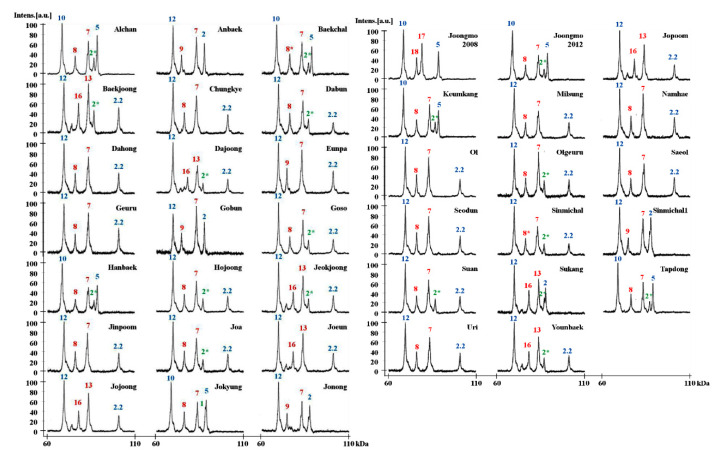
HMW-GS profiles of 38 Korean wheat cultivars of individual subunits determined by MALDI-TOF-MS. Subunits corresponding to chromosomes 1A, 1B, and 1D are shown in green, red, and blue, respectively.

**Table 1 molecules-25-04347-t001:** Major factors influencing resolution of high-molecular-weight glutenin subunits (HMW-GS) by MALDI-TOF-MS.

Factors	Parameters
**Alkylating reagent**	DTT
	DTT, 4-vinylpyridine
**Solvent components**	0.1% TFA in 30% ACN
	0.1% TFA in 50% ACN
	SA dissolved in 0.1% TFA in 30% ACN
	SA dissolved in 0.1% TFA in 50% ACN
**Dilution volume**	10 μL
	50 μL
	100 μL
**Matrix II**	SA dissolved in 30% ACN
	SA dissolved in 50% ACN
	SA dissolved in 0.1% TFA in 30% ACN
	SA dissolved in 0.1% TFA in 50% ACN

**Table 2 molecules-25-04347-t002:** Reproducibility of molecular weights for HMW-GS determined by MALDI-TOF-MS in standard wheat cultivars.

	HMW-GS				MW (Da) ^1^					RSD (%) ^2^				
	*Glu-A1*	*Glu-B1*	*Glu-D1*	Reference	*Glu-A1*	*Glu-B1*		*Glu-D1*		*Glu-A1*	*Glu-B1*		*Glu-D1*	
Cultivar Name	x	x + y	x + y		x	x	y	x	y	x	x	y	x	y
APEXAL	N	7 + 9	5 + 10	Branlard et al. [44]		82,831	74,264	88,458	68,388		0.036	0.054	0.050	0.032
BRIMSTONE	N	6 + 8*	2 + 12	Liu et al. [50]		86,840	75,993	87,431	69,404		0.076	0.083	0.014	0.071
CAPPELLE-DESPREZ	N	7	2 + 12	Liu et al. [50]		83,416		87,364	69,390		0.072		0.050	0.054
CHEYENNE	2*	7 + 9	5 + 10	Dupont et al. [46]	86,556	82,824	74,295	88,450	68,400	0.036	0.076	0.059	0.078	0.073
CHINESE SPRING	N	7 + 8	2 + 12	Liu et al. [50]		83,488	75,886	87,493	69,410		0.013	0.025	0.020	0.009
CLEMENT	N	6 + 8*	2 + 12	Liu et al. [50]		86,828	75,981	87,379	69,414		0.091	0.066	0.037	0.069
GLENLEA	2*	7^oe^ + 8*	5 + 10	Naeem and Sapirstein [21]	86,591	83,402	75,878	88,477	68,353	0.025	0.068	0.050	0.031	0.034
HANNO	1	14 + 15	5 + 10	Gao et al. [20]	87,986	82,786	75,526	88,459	68,346	0.030	0.039	0.035	0.068	0.021
IMBROS	1	14 + 15	2 + 12	Kazman et al. [47]	87,902	82,767	75,552	87,423	69,404	0.039	0.035	0.033	0.025	0.037
INSIGNIA	1	20 + 20	5 + 10	Branlard et al. [44]	87,982	84,014	75,859	88,446	68,383	0.015	0.078	0.057	0.063	0.068
MAGNIF27	2*	13 + 16	2 + 12	Branlard et al. [44]	86,661	83,475	77,790	87,413	69,409	0.071	0.061	0.084	0.081	0.070
NANBU-KOMUGI	1	7 + 8	4 + 12	Liu et al. [50]	87,893	83,437	75,876	85,873	69,393	0.060	0.022	0.030	0.062	0.043
NEEPAWA	2*	7 + 9	5 + 10	Liu et al. [50]	86,608	82,881	74,274	88,506	68,388	0.068	0.028	0.044	0.055	0.072
NORIN 61	2*	7 + 8	2.2 + 12	Hua et al. [51]	86,572	82,842	75,869	100,811	69,362	0.042	0.067	0.077	0.044	0.065
OPATA	2*	13 + 16	2 + 12	Liu et al. [50]	86,555	83,466	77,824	87,371	69,425	0.036	0.055	0.072	0.076	0.025
ORCA	N	7	2 + 12	Liu et al. [50]		82,823		87,331	69,393		0.055		0.068	0.030
OREPI	1	7	4 + 12	Branlard et al. [44]	87,949	83,483		86,380	69,433	0.044	0.031		0.066	0.045
PETREL	N	7	5 + 10	Liu et al. [50]		83,405		88,432	68,339		0.052		0.020	0.078
SOISSONS	2*	7 + 8	5 + 10	Liu et al. [50]	86,601	82,803	75,865	88,457	68,337	0.048	0.069	0.059	0.032	0.074
THESEE	N	6 + 8*	2 + 12	Liu et al. [50]		86,861	76,017	87,381	69,439		0.048	0.046	0.046	0.036
SUKANG	2*	13 + 16	2 + 12	Park et al. [49]	86,623	83,448	77,794	87,370	69,422	0.092	0.058	0.044	0.053	0.088
MANITAL	2*	17 + 18	2 + 12	Liu et al. [50]	86,598	79,027	75,960	87,349	69,381	0.058	0.036	0.050	0.046	0.070
BAEKJOONG	2*	13 + 16	2.2 + 12	Park et al. [49]	86,571	83,480	77,791	100,809	69,444	0.079	0.057	0.052	0.063	0.056
IT166460	1	7^oe^ + 8*	2 + 12	Cho et al. [45]	87,909	83,534	75,961	87,450	69,453	0.037	0.018	0.039	0.030	0.019

^1^ Mean average of molecular weights determined from at least 5 analyses. ^2^ Relative standard deviation.

**Table 3 molecules-25-04347-t003:** Comparison of molecular weights of HMW-GS predicted from mature proteins deduced from gene sequences with actual molecular weights determined by MALDI-TOF-MS in standard cultivars.

Locus		HMW-GS (Accesion Number)	MW of Mature Protein (Da) ^1^	MW by MALDI-TOF-MS (Da) ^2^	Cultivars	Difference (Da)	Error (%)
**Glu-A1**	**x**	**1**	**87,679**	**87,565**	**Hanno**	**−114**	−0.13
		**(X61009)**		87,481	Imbros	−198	−0.23
				87,561	Insignia	−118	−0.13
				87,472	Nanbu-Komugi	−207	−0.24
				87,528	Orepi	−151	−0.17
				87,488	IT166460	−191	−0.22
		**2***	86,335	86,135	Cheyenne	−200	−0.23
		**(M22208)**		86,170	Glenlea	−165	−0.19
				86,240	Magnif 27	−95	−0.11
				86,187	Neepawa	−148	−0.17
				86,151	Norin 61	−184	−0.21
				86,134	Opata	−201	−0.23
				86,180	Soissons	−155	−0.18
				86,202	Sukang	−133	−0.15
				86,177	Manital	−158	−0.18
				86,150	Baekjoong	−185	−0.21
**Glu-B1**	**x**	**6**	86,523	86,419	Brimstone	−104	−0.12
		**(KX454509)**		86,407	Clement	−116	−0.13
				86,440	Thesee	−83	−0.10
		**7 (group 1)**	82,527	82,410	Apexal	−117	−0.14
		**(X13927)**		82,403	Cheyenne	−124	−0.15
				82,460	Neepawa	−67	−0.08
				82,421	Norin 61	−106	−0.13
				82,402	Orca	−125	−0.15
				82,382	Soissons	−145	−0.18
		**7 (group 2)**	82,527	82,995	Cappelle-Desprez	468	0.57
		**(X13927)**		83,067	Chinese Spring	540	0.65
				83,016	Nanbu-Komugi	489	0.59
				83,062	Orepi	535	0.65
				82,984	Petrel	457	0.55
		**7OE**	83,122	82,981	Glenlea	−141	−0.17
		**(EU157184)**		83,113	IT166460	−9	−0.01
		**13**	83,209	83,054	Magnif 27	−155	−0.19
		**(EF540764)**		83,045	Opata	−164	−0.20
				83,027	Sukang	−182	−0.22
				83,059	Baekjoong	−150	−0.18
		**14**	82,343	82,365	Hanno	22	0.03
		**(KF733216)**		82,346	Imbros	3	0.00
		**17**	77,960	78,606	Manital	646	0.83
		**(AB263219)**					
		**20**	83,895	83,803	Insignia	−92	−0.11
		**(AJ437000)**					
	**y**	**8**	75,159	75,150	Chinese Spring	−9	−0.01
		**(AY245797)**		75,140	Nanbu-Komugi	−19	−0.03
				75,133	Norin 61	−26	−0.03
				75,133	Soissons	−26	−0.03
		**8***	N/A	75,257	Brimstone		
				75,245	Clement		
				75,142	Glenlea		
				75,281	Thesee		
				75,225	IT166460		
		**9**	73,517	73,528	Apexal	11	0.01
		**(X61026)**		73,559	Cheyenne	42	0.06
				73,538	Neepawa	21	0.03
		**15**	74,738	74,790	Hanno	52	0.07
		**(KF733215)**		74,816	Imbros	78	0.10
		**16**	77,283	77,054	Magnif 27	−229	−0.30
		**(EF540765)**		77,088	Opata	−195	−0.25
				77,058	Sukang	−225	−0.29
				77,055	Baekjoong	−228	−0.29
		**18**	75,187	75,224	Manital	37	0.05
		**(KF430649)**					
		**20** **(LN828972)**	75,148	75,123	Insignia	−25	−0.03

**Glu-D1**	**x**	**2**	87,007	87,010	Brimstone	3	0.00
		**(X03346)**		86,943	Cappelle-Desprez	−64	−0.07
				87,072	Chinese Spring	65	0.07
				86,958	Clement	−49	−0.06
				87,002	Imbros	−5	−0.01
				86,992	Magnif 27	−15	−0.02
				86,950	Opata	−57	−0.07
				86,910	Orca	−97	−0.11
				86,960	Thesee	−47	−0.05
				86,949	Suknag	−58	−0.07
				86,928	Manital	−79	−0.09
				87,029	IT166460	22	0.03
		**2.2**	10,0886	10,0390	Norin 61	−496	−0.49
		**(AY159367)**		10,0388	Baekjoong	−498	−0.49
		**4**	N/A	85,452	Nanbu-Komugi		
				85,959	Orepi		
		**5**	88,126	87,932	Apexal	−194	−0.22
		**(X12928)**		87,924	Cheyenne	−202	−0.23
				87,951	Glenlea	−175	−0.20
				87,933	Hanno	−193	−0.22
				87,920	Insignia	−206	−0.23
				87,980	Neepawa	−146	−0.17
				87,906	Petrel	−220	−0.25
				87,931	Soissons	−195	−0.22
	**y**	**10**	67,475	67,652	Apexal	177	0.26
		**(X12929)**		67,664	Cheyenne	189	0.28
				67,617	Glenlea	142	0.21
				67,610	Hanno	135	0.20
				67,647	Insignia	172	0.25
				67,652	Neepawa	177	0.26
				67,603	Petrel	128	0.19
				67,601	Soissons	126	0.19
		**12**	68,713	68,668	Brimstone	−45	−0.07
		**(X03041)**		68,654	Cappelle-Desprez	−59	−0.09
				68,674	Chinese Spring	−39	−0.06
				68,678	Clement	−35	−0.05
				68,668	Imbros	−45	−0.07
				68,673	Magnif 27	−40	−0.06
				68,637	Nanbu-Komugi	−76	−0.11
				68,626	Norin 61	−87	−0.13
				68,689	Opata	−24	−0.03
				68,657	Orca	−56	−0.08
				68,697	Orepi	−16	−0.02
				68,703	Thesee	−10	−0.01
				68,686	Sukang	−27	−0.04
				68,645	Manital	−68	−0.10
				68,708	Baekjoong	−5	−0.01
				68,717	IT166460	4	0.01

^1^ MW of mature proteins deduced from gene sequences was determined using the Expasy server (https://web.expasy.org/compute_pi/). N/A indicates that the MW could not be determined because gene sequences were not available for the subunit, ^2^ Molecular weight excluding 4-vp. Most x-type and y-type HMW-GS have 4 and 7 cysteine residues, respectively, except for 1Bx20 and 1Dx5 that have 2 and 5 cysteine residues, respectively.

**Table 4 molecules-25-04347-t004:** Reproducibility of molecular weights (MW) for HMW-GS in Korean wheat cultivars by MALDI-TOF-MS.

	HMW-GS			MW (Da) ^1^					RSD(%) ^2^				
	*Glu-A1*	*Glu-B1*	*Glu-D1*	*Glu-A1*	*Glu-B1*		*Glu-D1*		*Glu-A1*	*Glu-B1*		*Glu-D1*	
Specific Name	x	x + y	x + y	x	x	y	x	y	x	x	y	x	y
Alchan	2*	7 + 8	5 + 10	86,618	83,457	75,924	88,493	68,402	0.018	0.036	0.039	0.028	0.040
Anbaek	N	7 + 9	2 + 12		82,885	74,323	87,424	69,452		0.037	0.032	0.024	0.035
Baekchal	2*	7 + 8*	5 + 10	86,606	82,859	75,874	88,475	68,416	0.021	0.027	0.048	0.023	0.053
Baekjoong	2*	13 + 16	2.2 + 12	86,595	83,478	77,808	100,834	69,443	0.026	0.060	0.096	0.026	0.093
Chungkye	N	7 + 8	2.2 + 12		82,905	75,907	100,854	69,427		0.034	0.016	0.013	0.053
Dabun	2*	7 + 8	2.2 + 12	86,582	83,437	75,889	100,834	69,402	0.031	0.034	0.019	0.019	0.039
Dahong	N	7 + 8	2.2 + 12		83,444	75,907	100,821	69,415		0.033	0.031	0.017	0.080
Dajoong	2*	13 + 16	2.2 + 12	86,634	83,505	77,858	100,834	69,456	0.025	0.040	0.054	0.015	0.046
Eunpa	N	7 + 9	2.2 + 12		82,851	74,312	100,855	69,415		0.039	0.077	0.032	0.047
Geuru	N	7 + 8	2.2 + 12		83,445	75,884	100,828	69,417		0.032	0.037	0.031	0.039
Gobun	N	7 + 9	2 + 12		82,903	74,353	87,427	69,427		0.042	0.021	0.022	0.054
Goso	2*	7 + 8	2.2 + 12	86,601	83,429	75,922	100,838	69,467	0.017	0.035	0.045	0.033	0.025
Hanbaek	2*	7 + 8	5 + 10	86,661	83,492	75,922	88,491	68,408	0.047	0.030	0.038	0.048	0.052
Hojoong	2*	7 + 8	2.2 + 12	86,631	82,886	75,926	100,830	69,445	0.054	0.046	0.011	0.015	0.045
Jeokjoong	2*	13 + 16	2.2 + 12	86,659	83,488	77,826	100,826	69,447	0.056	0.055	0.073	0.033	0.081
Jinpoom	N	7 + 8	2.2 + 12		82,907	75,926	100,842	69,449		0.035	0.031	0.024	0.029
Joa	2*	7 + 8	2.2 + 12	86,617	82,913	75,932	100,852	69,462	0.032	0.053	0.036	0.020	0.046
Joeun	N	13 + 16	2.2 + 12		83,505	77,876	100,835	69,447		0.032	0.045	0.028	0.043
Jojoong	N	13 + 16	2.2 + 12		83,504	77,858	100,836	69,446		0.033	0.038	0.018	0.039
Jokyung	1	7 + 8	5 + 10	87,970	83,433	75,902	88,473	68,361	0.027	0.037	0.048	0.012	0.033
Jonong	N	7 + 9	2 + 12		82,874	74,308	87,421	69,437		0.050	0.035	0.028	0.017
Joonmo2008	N	17 + 18	5 + 10		79,062	75,969	88,480	68,374		0.030	0.047	0.035	0.039
Joongmo2012	2*	7 + 8	5 + 10	86,632	83,442	75,896	88,484	68,411	0.044	0.041	0.046	0.022	0.050
Jopoom	N	13 + 16	2.2 + 12		83,490	77,858	100,865	69,440		0.044	0.064	0.024	0.038
Keumkang	2*	7 + 8	5 + 10	86,580	83,448	75,907	88,500	68,395	0.033	0.052	0.046	0.030	0.064
Milsung	N	7 + 8	2.2 + 12		83,448	75,911	100,857	69,455		0.025	0.051	0.035	0.031
Namhae	N	7 + 8	2.2 + 12		82,909	75,911	100,848	69,463		0.028	0.040	0.016	0.029
Ol	N	7 + 8	2.2 + 12		82,868	75,883	100,846	69,426		0.028	0.021	0.037	0.037
Olgeuru	2*	7 + 8	2.2 + 12	86,672	83,461	75,927	100,848	69,458	0.031	0.031	0.060	0.022	0.040
Saeol	N	7 + 8	2.2 + 12		83,434	75,924	100,846	69,441		0.042	0.069	0.029	0.043
Seodun	N	7 + 8	2.2 + 12		82,898	75,936	100,851	69,451		0.056	0.018	0.023	0.045
Sinmichal	2*	7 + 8*	2.2 + 12	86,640	82,877	75,908	100,852	69,425	0.054	0.028	0.034	0.017	0.038
Sinmichal1	N	7 + 9	2 + 12		82,863	74,328	87,389	69,424		0.042	0.031	0.042	0.053
Suan	2*	7 + 8	2.2 + 12	86,616	82,858	75,918	100,853	69,418	0.017	0.028	0.034	0.017	0.020
Sukang	2*	13 + 16	2 + 12	86,660	83,522	77,831	87,410	69,420	0.020	0.033	0.033	0.034	0.015
Tapdong	2*	7 + 8	5 + 10	86,628	82,884	75,942	88,503	68,384	0.022	0.019	0.036	0.032	0.038
Uri	N	7 + 8	2.2 + 12		83,461	75,923	100,819	69,438		0.035	0.036	0.025	0.053
Younbaek	2*	13 + 16	2.2 + 12	86,644	83,476	77,861	100,838	69,433	0.024	0.026	0.046	0.023	0.055

^1^ Mean averages of molecular weights determined from at least 5 analyses of each cultivar. ^2^ Relative standard deviation.

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
