# Peer review of "Development of an Optimized MALDI-TOF-MS Method for High-Throughput Identification of High-Molecular-Weight Glutenin Subunits in Wheat"

_molecules, 2020, doi:10.3390/molecules25184347_

Round 1

Reviewer 1 Report

The manuscript entitled “Development of an optimized MALDI-TOF-MS method for high-throughput identification of high-molecular-weight glutenin subunits in wheat” by Lee and colleagues describes the optimization of a protocol for a fast and precise identification of high-molecular-weight glutenin subunits in wheat. In addition, the optimized protocol has been validated through the screening of a wheat Korean collection.

The ms is well written, the results are well described and discussed. The major novelty is the discrimination of the subunits 1Bx7and 1Bx7OE. However, this study confirmed that the use of MALDI-TOF-MS method has yet some limitations to distinguish subunits with very similar molecular weights.

I am curious to know if this method is able to distinguish the 1DX5 subunits with and without the additional cysteine. I suggest to add this analysis in the manuscript. The two subunits have the same mobility in SDS-PAGE.

Line 167. The authors claimed that the sequence of By20 is not present in the databases. It is not correct, because the sequences of Bx20 and By20 were isolated by Santagati et al 2016 Journal of Mass Spectrometry DOI: 10.1002/jms.3776 (accession n. LN828972.1).

Author Response

Reviewer #1

Reviewer Comment 1:
I am curious to know if this method is able to distinguish the 1DX5 subunits with and without the additional cysteine. I suggest to add this analysis in the manuscript. The two subunits have the same mobility in SDS-PAGE.

Response:
Assuming that the only difference between the two subunits is the substitution of one cysteine with a serine as in the mutant Dx5 described by Wang et al., 2017, JAFC 65:6264-6273, it would not be possible to distinguish the two proteins by MALDI-TOF-MS. The MWs of the two proteins would differ by only 105 Da because of the alkylation of the extra cysteine residue.

We also performed a BLASTp search of NCBI with the Dx5 sequence P10388 to identify closely related HMW-GS with four cysteines instead of five. This search revealed a sequence from A. tauschii (AAV52919) that differs from P10388 by a six amino acid insertion as well as five single amino acid changes, including the substitution of a serine for a cysteine. The predicted MW of P10388 after alkylation would be 88,653 while that of AAV52919 would be 87,042. With a difference of 1611 Da, these would be easily distinguished by MALDI-TOF-MS. We were unable to find any other HMW-GS sequences with four cysteines that were more closely related to P10388.

Reviewer Comment 2:
Line 167. The authors claimed that the sequence of By20 is not present in the databases. It is not correct, because the sequences of Bx20 and By20 were isolated by Santagati et al 2016 Journal of Mass Spectrometry DOI: 10.1002/jms.3776 (accession n. LN828972.1).

Response:
Thank you for bringing this paper to our attention. We have added this sequence and the predicted MW to Table 3. We have also modified line 164 of the manuscript accordingly.

Reviewer 2 Report

The manuscript shows a novel and interesting approach for the determination of high-molecular-weight glutenin subunits in wheat samples. The method was developed very rigorously. It is clearly shown the benefit of using MALDI-ToF-MS capability for the goal of this work.

Please find below some comments:

In line 289: I think that 30mg should be corrected to 30 mg

In line 311: 96 experiments in total were carried out. Is was considering to run a design of experiments to optimize the number of runs and most probably reduce the number of experiments?

In lines 220, 222, 224: Please keep the space among numbers in the gene sequences.

In line 243, 244: It is mentioned that red and blue boxes indicate subunits that can not be distinguished by MALDI-TOF-MS because their molecular weights differ by less than 300 Da. Is that information important? It would be good to propose an alternative way to solve this problem.

Author Response

Reviewer #2

We have made minor corrections to the text as suggested by the reviewer.

Reviewer Comment 1:
In line 311: 96 experiments in total were carried out. Is was considering to run a design of experiments to optimize the number of runs and most probably reduce the number of experiments?
Response:
The experimental design to optimize the separation of HMW-GS by MALDI-TOF-MS involved 96 experiments in which different parameters were evaluated. Conditions that yielded the best spectra are shown in Figure 1 and were used for the analysis of both standard cultivars and Korean cultivars. We have clarified this on lines 97 and 98 and have removed this statement from line 311.

Reviewer Comment 2:
In line 243, 244: It is mentioned that red and blue boxes indicate subunits that can not be distinguished by MALDI-TOF-MS because their molecular weights differ by less than 300 Da. Is that information important? It would be good to propose an alternative way to solve this problem.
Response:
We have modified lines 244-245 to indicate that the subunits shown in boxes that were not distinguished by MALDI-TOF-MS could be differentiated by RP-HPLC.